# Epidemiology of Systemic Mycoses in the COVID-19 Pandemic

**DOI:** 10.3390/jof7070556

**Published:** 2021-07-13

**Authors:** María Guadalupe Frías-De-León, Rodolfo Pinto-Almazán, Rigoberto Hernández-Castro, Eduardo García-Salazar, Patricia Meza-Meneses, Carmen Rodríguez-Cerdeira, Roberto Arenas, Esther Conde-Cuevas, Gustavo Acosta-Altamirano, Erick Martínez-Herrera

**Affiliations:** 1Unidad de Investigación, Hospital Regional de Alta Especialidad de Ixtapaluca, Ciudad de México PC 56530, Estado de México, Mexico; magpefrias@gmail.com (M.G.F.-D.-L.); rodolfopintoalmazan@gmail.com (R.P.-A.); eduardogs_01@hotmail.com (E.G.-S.); mq9903@live.com.mx (G.A.-A.); 2Sección de Estudios de Posgrado e Investigación, Escuela Superior de Medicina, Instituto Politécnico Nacional, Plan de San Luis y Díaz Mirón s/n, Col. Casco de Santo Tomas, Alcaldía Miguel Hidalgo, Ciudad de México PC 11340, Estado de México, Mexico; 3Departamento de Ecología de Agentes Patógenos, Hospital General “Dr. Manuel Gea González”, Ciudad de México PC 14080, Estado de México, Mexico; rigo37@gmail.com; 4Maestría en Ciencias de la Salud, Escuela Superior de Medicina, Instituto Politécnico Nacional, Plan de San Luis y Díaz Mirón s/n, Col. Casco de Santo Tomas, Alcaldía Miguel Hidalgo, Ciudad de México PC 11340, Estado de México, Mexico; patricia_meza@ymail.com (P.M.-M.); condeesther999@gmail.com (E.C.-C.); 5Servicio de Infectología, Hospital Regional de Alta Especialidad de Ixtapaluca, Ciudad de México PC 56530, Estado de México, Mexico; 6Efficiency, Quality, and Costs in Health Services Research Group (EFISALUD), Galicia Sur Health Research Institute (IIS Galicia Sur), SERGAS-UVIGO, PC 36213 Vigo, Spain; carmencerdeira33@gmail.com (C.R.-C.); rarenas98@gmail.com (R.A.); 7Dermatology Department, Hospital Vithas Ntra. Sra. de Fátima and University of Vigo, PC 36206 Vigo, Spain; 8Campus Universitario, University of Vigo, PC 36310 Vigo, Spain; 9Sección de Micología, Hospital General “Dr. Manuel Gea González”, Ciudad de México PC 14080, Estado de México, Mexico

**Keywords:** co-infection, COVID-19, systemic mycoses, pneumocystosis, candidiasis, aspergillosis, mucormycosis, endemic mycosis

## Abstract

The physiopathologic characteristics of COVID-19 (high levels of inflammatory cytokines and T-cell reduction) promote fungal colonization and infection, which can go unnoticed because the symptoms in both diseases are very similar. The objective of this work was to study the current epidemiology of systemic mycosis in COVID-19 times. A literature search on the subject (January 2020–February 2021) was performed in PubMed, Embase, Cochrane Library, and LILACS without language restrictions. Demographic data, etiological agent, risk factors, diagnostic methods, antifungal treatment, and fatality rate were considered. Eighty nine publications were found on co-infection by COVID-19 and pneumocystosis, candidiasis, aspergillosis, mucormycosis, coccidioidomycosis, or histoplasmosis. In general, the co-infections occurred in males over the age of 40 with immunosuppression caused by various conditions. Several species were identified in candidiasis and aspergillosis co-infections. For diagnosis, diverse methods were used, from microbiological to molecular. Most patients received antifungals; however, the fatality rates were 11–100%. The latter may result because the clinical picture is usually attributed exclusively to SARS-CoV-2, preventing a clinical suspicion for mycosis. Diagnostic tests also have limitations beginning with sampling. Therefore, in the remainder of the pandemic, these diagnostic limitations must be overcome to achieve a better patient prognosis.

## 1. Introduction

SARS-CoV-2 is a betacoronavirus that causes the “2019 coronavirus disease” (COVID-19). Since the World Health Organization (WHO) declared a pandemic in March 2020, the world has been facing a health crisis that has involved significant challenges in diagnosing, treating, and preventing COVID-19 and its complications. As of 21 March 2021, it has affected 122,524,424 people in 222 countries resulting in the deaths of 2,703,620 individuals, showing an overall fatality rate of 2.2% [1]. The clinical course and progression of SARS-CoV-2 infection is variable and can lead to respiratory failure [2]. Clinical presentations of COVID-19 include fever (≥38 °C), cough, shortness of breath, loss of smell, chills, headache, and joint pain, among the most frequently reported. In most cases, the disease is asymptomatic or with mild to moderate symptomatology (80–85%) [3]. However, the disease can evolve to more severe clinical pictures in some patients, putting their lives at risk.

Reports indicate that the overall host state and the presence of comorbidities facilitate the spread of the virus and tropism of target organs with angiotensin-converting enzyme 2 (ACE2) receptors and increased production of interleukin (IL)-6, IL-1, and tumor necrosis factor (TNF)-α in severe cases [2]. In the case of the disease requiring hospitalization, it is characterized by pneumonia, lymphopenia, and cytokine release syndrome (CRS), which trigger an exaggerated immune response that causes damage at the local and systemic level. The previous occurs because, after the virus enters the cells and releases its genetic material (RNA), it is recognized by innate immunity receptors located intracellularly. Receptors such as the Toll 7 (TLR7), retinoic acid-inducible gene I (RIG-1), and melanoma differentiation-associated protein 5 (MDA-5), activate a signaling cascade, leading to the expression of interferon (IFN) type I (α and β), the objective of which is to interfere with viral replication. On the other hand, viral antigens can be processed by antigen-presenting cells through their major histicompatibility complex (MHC)-I to T cell receptor (TCR) of T CD8+ lymphocyte, which involves releasing their proteolytic enzymes (cytotoxicity). At the same time, the cytokine storm begins (an increase of IL-1B, IL-6, IL-7, IL-8, IL-9, IL-10, and TNF-α [2,3,4].

In mild pneumonia, patients may be asymptomatic or have at least one of the following symptoms: fever (constant or intermittent), dry cough, sore throat, headache, myalgia, tiredness, or diarrhea. However, they do not present dyspnea nor hypoxia [3]. Although in most patients (81%), the course of the disease is mild, some develop severe pneumonia, characterized by the acute respiratory distress syndrome (ARDS), refractory hypoxemia, and dyspnea. More than 50% experience lung damage radiologically characterized by the presence of bilateral ground-glass opacities. In some cases, patients also develop damage and dysfunction of the extrapulmonary system, such as alterations in the hematological and digestive system, risk of sepsis, and septic shock, with a high mortality rate [4]. Some of these patients present patchy bilateral opacities in the lungs, leukopenia or lymphopenia, and increased serum levels of alanine aminotransferase (ALT), aspartate transaminase (AST), lactate dehydrogenase (LDH), creatine kinase myocardial band (CK-MB), C-reactive protein (CRP), and erythrocyte sedimentation rate (ESR) [5].

Approximately 5 to 30% of COVID-19 patients have been reported to become critically ill, require mechanical ventilation, and admission to the Intensive Care Unit (ICU) [2,3], which constitutes a risk factor for the development of co-infections with other bacterial, viral, or fungal pathogens. SARS-CoV-2 infection is characterized by a cytokine storm involving increased inflammatory proteins (TNF-α, IL-6, IL-2R) and decreased anti-inflammatory proteins. The latter contributes to lung pathology and lymphopenia development (drastic decrease in T cells, CD4+ and CD8+) in patients admitted to the ICU [5]. Lymphopenia in patients with severe COVID-19 is a clinical situation that increases the risk of severe fungal infections by different fungal genera, such as *Pneumocystis*, *Candida*, and *Aspergillus*, as T cells are the second most crucial line of defense against mycoses [6,7].

Fungal co-infections in COVID-19 patients is not a surprising situation. There are precedents that viral pneumonia, such as H1N1 and H7N9 influenza, can have co-infections caused by fungi and bacteria, and even other viruses [8,9,10,11,12,13,14,15,16,17]. During the outbreak of severe acute respiratory syndrome (SARS) detected in Guangdong, China, in November 2002, that subsequently spread to more than 30 countries, a series of autopsy cases were reported, which evidenced invasive fungal infection compatible with invasive aspergillosis in 10% of cases [18]. Interestingly, all infected patients were treated with high doses of corticosteroids, possibly resulting in immunosuppression that facilitated the establishment of mycoses. Another autopsy study of individuals who died of SARS confirmed fungal co-infection with the isolation of *Aspergillus* sp. and *Mucor* sp. in the upper respiratory tract and lungs. A case with multiple *Aspergillus* abscesses in different organs was also reported [19]. With this background, the co-infection scenario is expected to be similar in SARS-CoV-2 pneumonia.

This work analyzes the epidemiology of fungal co-infections in COVID-19 patients based on the review of publications on the subject during the first year of the pandemic.

## 2. Materials and Methods

A thorough search of publications on the subject was performed from January 1st, 2020, to 28 February 2021. The search was conducted on PubMed, Embase, Cochrane Library, and LILACS. The search words used were “fungi” OR “fungus” OR “fungal infection” OR “invasive fungal diseases” OR “pneumocystis” OR “candidiasis” OR “aspergillosis” OR “mucor” OR “histoplasmosis” OR “coccidioidomycosis” OR “co-infection” OR “secondary infection” AND “COVID-19” OR “SARS-CoV-2” OR “2019-nCoV” OR “2019 novel coronavirus” without language restrictions. Cases caused by *Cryptococcus*, *Fusarium*, *Saccharomyces*, and other fungi were excluded from the study due to the low number of reports. The review was performed based on the preferred reporting items for systematic reviews and meta-analyses (PRISMA) (Figure 1).

## 3. Pneumonia by *Pneumocystis jirovecii* and COVID-19

During the COVID-19 pandemic, 13 publications have reported a total of 24 confirmed cases of co-infection with SARS-CoV-2 and *P. jirovecii* (Table 1). This co-infection has occurred in both men and women, in the age range of 11–83 years, predominantly in males over 40 years of age [20,21,22,23,24,25,26,27,28,29,30,31,32]. The main risk factor for developing co-infection with *Pneumocystis* was HIV infection with low CD4+ count, followed by immunosuppressive treatments, lymphopenia, and autoimmune disease (anti-melanoma differentiation-associated gene 5 juvenile dermatomyositis) [20,21,22,23,24,25,26,27,28,29,30,31,32]. Seven of these cases did not identify a typical risk factor for pneumocystosis [20]. The fungal presence was confirmed in 22 cases through different methods, such as PCR, high-performance sequencing, detection of β-D-glucan in serum, and staining techniques (Grocott and direct fluorescent antibody stain) [20,21,22,23,24,25,27,28,29,30,31,32]. Treatment of pneumocystosis in COVID-19 patients was trimethoprim-sulfamethoxazole [20,21,22,23,24,25,26,27,28,29,30,31,32]. However, one patient presented intolerance to this medication, so he was treated with clindamycin [26]. In five cases, prednisone/prednisolone was added to treatment with trimethoprim-sulfamethoxazole [21,24,27,28,32]. In one case, the patient was successfully treated with caspofungin acetate [23]. The fatality rate of COVID-19 and pneumocystosis was 33.3% [20,21,22,23,24,25,26,27,28,29,30,31,32].

## 4. Candidiasis and COVID-19

In the first year of the COVID-19 pandemic, 29 articles on co-infection by SARS-CoV-2 and *Candida* spp. have been published (Table 2). Co-infections by *Candida* have been observed in both women and men, but predominantly in men over the age of 40 [33,34,35,36,37,38,39,40,41,42,43,44,45,46,47,48,49,50,51,52,53,54,55,56,57,58,59,60,61]. The predisposing factors of candidiasis were multiple, including mechanical ventilation, placement of a central venous catheter, diabetes mellitus, antibiotics treatment, hospitalization time, anti-inflammatory treatment, and cancer [33,34,35,36,37,38,39,40,41,42,43,44,45,46,47,48,49,50,51,52,53,54,55,56,57,58,59,60,61]. Other less common risk factors were asthma, HIV, and surgery, among others [34,35,36,38,41,45,47,48,49,50,52,56,60]. The diagnosis was performed through culture, PCR, sequencing, and MALDI-TOF [33,34,35,36,37,39,40,41,42,43,44,45,46,47,48,49,50,51,52,53,54,55,56,57,58,59,60,61]. The most identified pathogen was *C. albicans*, followed by *C. glabrata*, *C. parapsilosis*, *C. tropicalis*, *C. auris*, *C. krusei*, *C. lusitaniae*, *C. inconspicua*, *C. dubliniensis*, *C. orthopsilosis*, *Candida* spp. and non-*albicans Candida* [33,34,35,36,37,38,39,40,41,42,43,44,45,46,47,48,49,50,51,52,53,54,55,56,57,58,59,60,61]. Patient treatment included azole antifungal agents (fluconazole, voriconazole, isavuconazole), echinocandins (caspofungin, anidulafungin, micafungin), and polyenes (amphotericin B, nystatin) [33,34,35,40,42,43,44,46,47,48,51,52,53,54,56,59]. It should be noted that nystatin was used for the treatment of a patient with oropharyngeal candidiasis [43]. Despite antifungal treatment, the fatality rate among patients with COVID-19 and candidiasis oscillated between 11 and 100% [33,34,35,36,37,38,39,40,41,42,43,44,45,46,47,48,49,50,51,52,53,54,55,56,57,58,59,60,61].

## 5. Aspergillosis and COVID-19

During the first year of the COVID-19 pandemic, 33 articles reporting aspergillosis co-infection in patients with COVID-19 have been published (Table 3). The most frequent causal agent was *A. fumigatus*, followed by *A. flavus*, *Aspergillus* spp. *A. niger*, *A. terreus*, *A. lentulus*, *A. nidulans*, *A. awamori*, *A. penicillioides*, and *A. citrinoterreus* [23,56,62,63,64,65,66,67,68,69,70,71,72,73,74,75,76,77,78,79,80,81,82,83,84,85,86,87,88,89,90,91,92]. The most used diagnostic methods were cultivation, galactomannan and (1,3)-β-D-glucan detection, MALDI-TOF, histopathology, serological tests, PCR, and sequencing [23,56,62,63,64,65,66,67,68,69,70,71,72,73,74,75,76,77,78,79,80,81,82,83,84,85,86,87,88,89,90,91,92]. It should be noted that in some of the *A. fumigatus* isolates, the *TR34/L98H* mutation in the Cyp51A gene was detected, which is associated with environmental azole resistance to antifungal agents [78,83]. Caspofungin, voriconazole, amphotericin B, anidulafungin, posaconazole, itraconazole, fluconazole or isavuconazole were used in antifungal treatment [23,56,62,63,64,66,67,69,70,71,72,74,76,77,78,79,80,81,82,84,85,86,87,88,89,90].

The fatality rate in cases of *Aspergillus* co-infection with SARS-CoV-2 was between 13% and 100% [23,56,62,63,64,65,66,67,68,69,70,71,72,73,74,75,76,77,78,79,80,81,82,83,84,85,86,87,88,89,90,91,92]. The affected patients were mainly men between 42 and 87 years of age [23,56,62,63,64,65,66,67,68,69,70,71,72,73,74,75,76,77,78,79,80,81,82,83,84,85,86,87,88,89,90,91,92]. The reported predisposing risk factors for the mycosis progression were diverse, including treatment with corticosteroids or immunosuppressants, hospitalization in the ICU, diabetes mellitus, hypertension, obesity, invasive mechanical ventilation, neoplasms, chronic obstructive pulmonary disease (COPD), asthma, and kidney disease [23,56,62,63,64,65,66,67,68,69,70,71,72,73,74,75,76,77,78,79,80,81,82,83,84,85,86,87,88,89,90,91,92].

## 6. Mucormycosis and COVID-19

In this study, nine publications of mucormycosis co-infection with COVID-19 were found (Table 4). Except for one case [93], all occurred in men in the age range of 24–86 years [93,94,95,96,97,98,99,100,101]. The predominant clinical form of mucormycosis was the rhino-orbital, followed by the pulmonary [93,94,95,96,97,98,99,100,101]. Only one case of each of the gastrointestinal [95] and rhinocerebral conditions was reported [93]. Among the main predisposing factors were diabetes mellitus, ketoacidosis, glucocorticoid use, or broad-spectrum antibiotics. Additional predisposing factors for lung mucormycosis were also reported, such as systemic high blood pressure, end-stage kidney disease, and ischemic cardiomyopathy [93,94,95,96,97,98,99,100,101]. The mycosis diagnosis was based mainly on histopathological analysis and culture [95,96,97,98,99,101]. In two cases, the clinical picture and the magnetic resonance imaging (MRI) were considered alone [93,94], whereas in one case, PCR and sequencing were used to detect and identify the pathogen [100]. The etiological agents in decreasing order were *Mucor* spp., *Rhizopus microsporus*, *Rhizopus* spp., and *Lichteimia* spp. [96,97,98,99,100]. In 36% of cases, the fungus was not identified [93,94,95,101]. Although the antifungal treatment included amphotericin B and isavuconazole, only 50% of patients showed clinical improvement [93,94,96,97,98,99,101]. The fatality rate was 50% [93,94,95,96,97,98,99,100,101].

## 7. Endemic Mycoses and COVID-19

Information on endemic fungal co-infections with SARS-CoV-2 is scarce. Only five cases have been published so far, two of COVID-19 and coccidioidomycosis in California, United States [102,103], and three of COVID-19 and histoplasmosis in Rio Grande, Southern Brazil, and Buenos Aires, Argentina [104,105,106] (Table 5). Coccidioidomycosis co-infections were caused by *C. immitis* [102,103]. They occurred in a man and a woman, both 48 years of age, with the main predisposing factor of residing in endemic areas [102,103]. The diagnosis was established with serological test results, and fluconazole was administered as an antifungal treatment. Both patients showed clinical improvement [102,103].

Histoplasmosis co-infections were caused by *H. capsulatum* and affected two women and one man in the range between 36 and 43 years of age [104,105,106]. Predisposing factors were HIV with CD4+ count < 200 cells/mm^3^ and residing in endemic areas [104,105,106]. The diagnosis in these cases was performed through staining methods (Gomori-Grocott, Wright, and Giemsa), blood culture, and antigen detection in urine and serum [104,105,106]. Antifungal treatment in COVID-19 patients with histoplasmosis was itraconazole and amphotericin B deoxycholate, which noticeably improved patients’ clinical state [104,105,106].

## 8. Future Challenges of Systemic Mycoses Co-Infections and COVID-19

To date, there is no specific treatment for patients with COVID-19. However, in severe cases, high-dose systemic glucocorticoids are administered as it has been observed that they improve patient survival. Broad-spectrum antibiotics are also used [96]. The latter, in addition to the specific physiopathologic characteristics of COVID-19, such as the cytokine storm and reduced T-cell levels, favor fungal co-infections [94,95,101,107]. However, reports of fungal co-infections in COVID-19 patients are scarce, probably because fungal lung infections, such as pneumocystosis, aspergillosis, histoplasmosis, and coccidioidomycosis, can be mistaken for SARS-CoV-2 infection.

Furthermore, they might be unnoticed because the symptoms in both diseases are very similar, including fever, dry cough, dyspnea, myalgia, and headache [7,103]. There are no pathognomonic radiographic or tomographic findings that can differentiate a COVID-19 pneumopathy from a fungal infection [108]. These facts undoubtedly limit proper patient therapeutic management, which can lead to fatal outcomes, as shown by the few reports we present in this work, where the fatality rate ranges from 11 to 100% (Table 1, Table 2, Table 3, Table 4 and Table 5). This situation poses three new challenges related to the diagnosis of fungal infections that will have to be overcome in the remainder of the pandemic.

In the first challenge, it is crucial that, in the case of evidence of fungal co-infection in COVID-19 patients, the clinical picture is not attributed exclusively to SARS-CoV-2 infection, particularly in severe cases. While many of the secondary infections in these patients are caused by bacteria or viruses, it is vital to not forget about fungi, particularly those with clinical pictures virtually indistinguishable from SARS-CoV-2 pneumonia, like *P. jirovecii*. In this sense, healthcare workers must consider the medical history and epidemiological data, as they may be key to establishing clinical suspicion and directing diagnosis by selecting appropriate laboratory or cabinet tests.

The second challenge is related to diagnostic tests. The conventional tests that identify a fungal pathogen and confirm a mycosis diagnosis have different limitations, such as the time required to obtain a result or the impossibility of determining the fungus at the species level. The latter is utterly important, especially in mycosis caused by different species, particularly from the genera *Candida*, *Aspergillus*, and the order Mucorales. It is worth mentioning that new species have been identified, and many species that had not previously been associated with infections in humans are now recognized as pathogenic within these fungal groups. Further, many of these new species may have different susceptibility to the antifungals of choice for mycosis treatment, for example, we can cite *C. auris* and *Aspergillus* section *Fumigati*. While it is true that the limitations of conventional diagnostic tests have been gradually overcome with the development and implementation of different tests, particularly molecular ones, not all tests are available in intrahospital laboratories due to the lack of adequate infrastructure [91].

Diagnosing mycoses with molecular techniques is often complicated because they require trained staff. It is also impractical to routinely analyze multiple samples. The use of PCR and other molecular methods provides limited detection and identification due to the presence of inhibitors in clinical samples. Such inhibitors can lead to underestimating fungus at different levels, from the extraction process to the amplification of nucleic acids. As for antigen detection, it is not always specific as there is potential antigenic cross-reactivity, which would prevent fungal identification [109].

The third challenge related to diagnostic tests is sample collection. For example, for COVID-19-associated pulmonary aspergillosis diagnosis, the sample must be collected from a bronchial washing that can generate aerosols representing a source of infection for healthcare personnel [62,75,110,111]. Another critical complication in diagnosis is the lack of algorithms and standardized diagnostic methods since the European Organization for Research and Treatment of Cancer (EORTC)/Mycoses Study Group (MSG) criteria are not specific for aspergillosis. For this reason, in an effort to establish a timely diagnosis, the AspICU algorithm for non-immunocompromised patients has been modified.

Overcoming these challenges will allow identification of cases of co-infections caused by SARS-CoV-2 and fungi, and enable definition of risk factors, affected populations, species distribution, and its antifungal profile.

Treatments reported in the revised literature fail to mention antifungal effectiveness as it is unknown whether the reported mortality was due to the stage of the disease per se or the fungal infection. In addition, treatment depends on the adequate diagnosis, which is complicated, as already mentioned. Thus, by not diagnosing the fungi properly, the correct treatment may not be given, and therefore it may not be effective.

It is important to mention that this work has limitations. This study did not include mycoses that were either reported just once, like fusariosis or caused by fungi that have rarely been associated with human infections, like *Saccharomyces cerevisiae*. Another limitation is that, unfortunately, not all fungal infections developed in COVID-19 patients have been reported in the literature. It should be noted that most references included in this study come from non-American countries, even though America has been the continent most affected by the pandemic. It is highly likely that fungal co-infections have occurred but have not been detected. It is our hope that the information presented in this study may serve as an alert for health workers to be aware of possible fungal co-infections in COVID-19 patients and the challenges posed by diagnosis. We also hope to remind them of the relevance of establishing acceptable infection control measures, prophylaxis, and adequate antifungal therapies.

## Figures and Tables

**Figure 1 jof-07-00556-f001:**
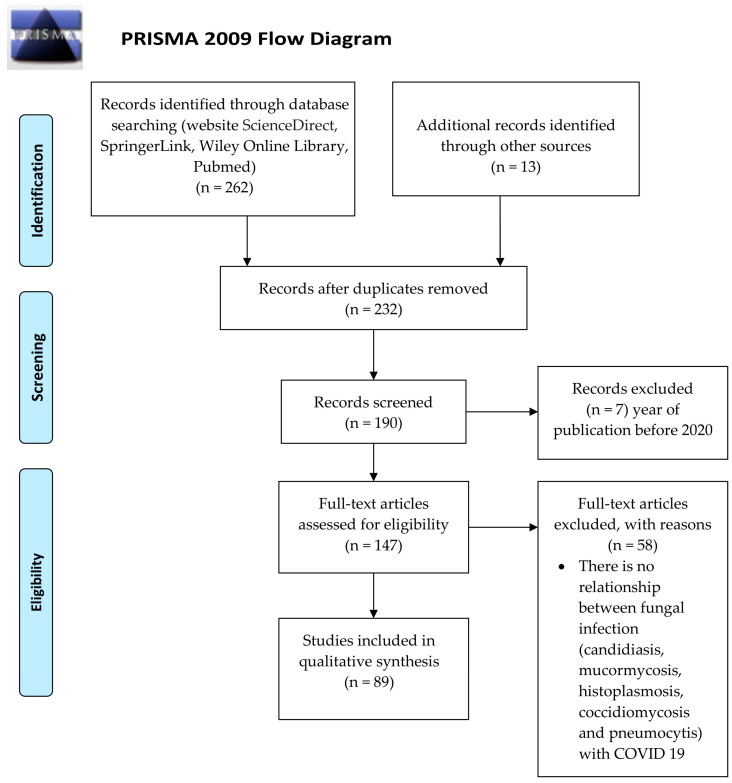
Flowchart of the different phases of the systematic review.

**Table 1 jof-07-00556-t001:** Cases of *COVID-19* and *Pneumocystis jirovecii* co-infection.

Number of Cases/Sex/Age (Years)/Country	Risk Factors	Diagnostic Method	Clinical Management	Fatality Rate (%)	Reference
10/8 male and 2 female/average between 46–68/France	Prolonged corticosteroid treatment in 3 patients	RT-qPCR; detection of *β*-D-glucan in serum	Cotrimoxazole in 4 patients, 6 patients were not treated	30.0	[20]
1/male/25/USA	HIV, CD4 = 32 cells/mm^3^	Detection of antigen in respiratory secretions by bronchoscopy	Trimethoprim-sulfamethoxazole, prednisone	0.0	[21]
2/female/78 and the other was not reported/France	Lymphocytopenia	qPCR in BAL	Specific anti-*Pneumocystis* treatment	100.0	[22]
1/female/72/China	30-year-old rheumatoid arthritis and glucocorticoid treatment	High troughput sequencing analysis	Caspofungin acetate, glucocorticoids	0.0	[23]
1/male/55/UK	HIV, CD4 422 cells/μL	PCR multiplex	Cotrimoxazole and prednisolone	0.0	[24]
1/male/65/Italy	Kidney transplant, immunosuppressive treatment	qPCR	Trimethoprim-sulfamethoxazole	100.0	[25]
2/ND/ND/China	HIV	ND	Clindamycin for one patient and trimethoprim-sulfamethoxazole for the other	0.0	[26]
1/female/46/Argentina	HIV, absolute CD4+ count 67 cells/μL	Grocott stain of a sputum sample	Ceftriaxone, azithromycin, trimethoprim-sulfamethoxazole, prednisone, and fluconazole	0.0	[27]
1/male/52/Germany	HIV, CD4+ 12 cells/μL viral load of 360,000 HIV-1 RNA copies/mL	*Pneumocystis* detection in BAL by a non-specified method, high LDH	Trimethoprim-sulfamethoxazol, prednisone	0.0	[28]
1/female/83/USA	Lymphocytopenia	qPCR of tracheal aspirate	Trimethoprim-sulfamethoxazole	0.0	[29]
1/male/65/France	Chemotherapy for chronic lymphocytic leukemia, lymphopenia	qPCR in BAL	Trimethoprim-sulfamethoxazole	0.0	[30]
1/female/11/Spain	anti-MDA5 JDM	RT-PCR	Trimethoprim-sulfamethoxazole	100.0	[31]
1/male/36/USA	HIV, absolute CD4 cell count was <10 cells/µL	DFA and PCR	Trimethoprim-sulfamethoxazole and prednisone	100.0	[32]

PCR: Polymerase Chain Reaction; RT-PCR: quantitative reverse transcription PCR; BAL: Bronchoalveolar. Lavage; qPCR: Quantitative Polymerase Chain Reaction; LDH: Lactate dehydrogenase; Anti-MDA5 JDM: Anti-melanoma differentiation-associated gene 5 juvenile dermatomyositis; DFA: direct fluorescent antibody stain; ND: no data.

**Table 2 jof-07-00556-t002:** Cases of COVID-19 and candidiasis co-infection.

Number of Cases/Sex/Age (Years)/Country	Etiological Agent	Risk Factors	Diagnostic Method	Clinical Management	Fatality Rate (%)	Reference
1/male/72/Austria	*C. glabrata*	Mechanical ventilation, central venous catheter, and hospitalization time	Culture	Caspofungin	0.0	[33]
11/7 male and 4 female/average 59/Brazil	*C. albicans, C. glabrata* and *C. tropicalis*	DM, central venous catheter, antibiotics treatment, and HIV	Culture	Fluconazole, anidulafungin, voriconazole, and amphotericin B deoxycholate	72.7	[34]
41/21 male and 20 female/average 62/Brazil	*C. albicans, C. tropicalis, C. parapsilosis* and *C. glabrata*	DM, mechanical ventilation, central venous catheter, surgery, hospitalization time, and hypotensive	Culture	Anidulafungin and fluconazole	61.0	[35]
60/32 male and 28 female/average 51/China	*Candida* spp.	ND	Real-time PCR	ND	ND	[36]
4/ND/ND/China	*C. albicans*, and *C. glabrata*	DM, mechanical ventilation, septic shock, acute respiratory and acute renal injury, and glucocorticoid treatment	Culture	ND	ND	[37]
2/ND/ND/China	*C. albicans*	DM, central venous catheter, peripherally inserted central catheter, glucocorticoid treatment, antibiotics treatment, and hematological disease	ND	ND	ND	[38]
6/3 male and 3 female/average 62/China	*C. albicans, C. parapsilosis C. lusitaniae* and *C. tropicalis*	DM, bacterial co-infection, higher white blood cell, and neutrophil counts, and higher levels of D-dimer, IL-6, IL-10, reactive protein-c, and procalcitonin	Culture and MALDI-TOF	ND	ND	[39]
20/13 male and 7 female/average 63/Colombia	*C. auris*, *C. albicans*, *C. tropicalis*, *C*. *parapsilosis* C. o*rthopsilosis*, C. *glabrata*	Mechanical ventilation, invasive hemodynamic support, prolonged stay in the ICU, DM, cancer, antibiotics treatment, and steroids	Culture and MALDI-TOF	Fluconazole, caspofungin, and voriconazole	60.0	[40]
5/ND/ND/Egypt	*C. albicans* and *C. glabrata*	Antibiotics treatment, anticoagulants, mechanical ventilation, oxygen therapy, ARDS, and renal injury	Culture	ND	ND	[41]
15/11 male and 4 female/average 63/India	*C. auris, C albicans, C. tropicalis*, and *C. krusei*	Mechanical ventilation, prolonged hospitalization time, DM, central lines and urinary catheters, and asthma	Culture, MALDI-TOF, and sequencing	Amphotericin B and micafungin	53.3	[42]
53/female 30 and male 23/average 63.1/Iran	*C. albicans*, *C. glabrata, C. dubliniensis*, *C. parapsilosis, C. tropicalis, and C. krusei*	Broad-spectrum antibiotics treatment, corticosteroid treatment, mechanical ventilation, and ICU stay period	PCR and sequencing	Fluconazole, nystatin, and caspofungin	ND	[43]
3/male/average 67.6/Italy	*C. albicans, C. parapsilosis* and *C. tropicalis*	Central venous catheter, parental nutrition, antibiotics treatment, and steroids treatment	Culture	Caspofungin, and fluconazole	0.0	[44]
36/ND/ND/Italy	*C. albicans, C. lusitaniae, C. glabrata, C. parapsilosis*, and *C. inconspicua*	ND	Culture and MALDI-TOF	ND	ND	[45]
7/ND/ND/Italy	*C. albicans, C. inconspicua* and *C. parapsilosis*	ICU stay period, mechanical ventilation, central venous catheter, antibiotics treatment, and corticosteroids treatment	Culture	Echinocandins	ND	[46]
6/ND/ND/Italy	*C. auris*	ICU length of stay, broad-spectrum antibiotics treatment, and asthma	Culture and MALDI-TOF, sequencing	Echinocandins	50.0	[47]
1/male/79/Italy	*C. glabrata*	Mechanical ventilation, antibiotics treatment, DM, and surgery	Culture and MALDI-TOF	Caspofungin	100.0	[48]
3/ND/ND/Italy	*C. albicans, C. parapsilosis* and *C. tropicalis*	Antibiotics treatment, HIV, cancer, DM, anti-inflammatory treatment, and hospital length of stay	Culture	ND	ND	[49]
21/16 male and 5 female/average 71/Italy	*C. albicans* and non *-albicans, Candida* spp.	Cancer, HIV, antibiotics treatment, parental nutrition, corticosteroid treatment, DM, ICU length of stay, central venous catheter, and surgery	Culture	ND	57.1	[50]
14/8 male and 6 female/average 72/Lebanon	*C. auris*	Cancer, ICU length of stay, mechanical ventilation, urinary catheter, central venous catheter, broad-spectrum antibiotics treatment, and steroids treatment	Culture and MALDI-TOF	Caspofungin and anidulafungin	35.7	[51]
12/10 male and 2 female/average 55/Mexico	*C. auris* and *C. glabrata*	Mechanical ventilation, peripherally inserted central lines, urinary catheter, asthma, steroids treatment, and prolonged ICU stay	Culture-MALDI-TOF, and sequencing	Isavuconazole, anidulafungin, caspofungin, amphotericin B, and voriconazole	83.3	[52]
5/male/average 59/Oman	*C. albicans, C. glabrata and C tropicalis*	Mechanical ventilation, ICU prolonged length of stay, broad-spectrum antibiotics treatment, and central line catheter	Culture and MALDI-TOF	Amphotericin B, caspofungin, and voriconazole	60.0	[53]
1/male/53/Spain	*C. albicans*	Mechanical ventilation, corticosteroid treatment, broad-spectrum antibiotics treatment, and central venous catheter	Culture	Fluconazole	0.0	[54]
3/ND/ND/UK	*C. albicans*	Central line, mechanical ventilation, immunomodulatory therapy, and broad-spectrum antibiotics treatment	Culture and MALDI-TOF	ND	ND	[55]
17/Ratio male:female was 2:1/aveage 58/UK	*C. albicans* and *C. parapsilosis*	Cancer, corticosteroid treatment, ventilation support, asthma, DM, and central venous catheter	Culture and MALDI-TOF	Liposomal amphotericin B, fluconazole, caspofungin, and voriconazole	38.5	[56]
35/21 male and 14 female/average 69/USA	*C. auris*	Central venous catheter, mechanical ventilator, urinary catheter, DM, cancer, nasogastric and gastric tube	Culture	ND	ND	[57]
1/male/54/USA	*C. albicans*	Cancer, and mechanical ventilation	Culture	ND	ND	[58]
8/4 male and 4 female/average 63/USA	*C. albicans, C. glabrata, C. parapsilosis* and *C. tropicalis*	ICU length of stay, mechanical ventilation, and central venous catheter	Culture and MALDI-TOF	Caspofungin and fluconazole	38.0	[59]
1/male/46/USA	*C. albicans*	Mechanical ventilation, cancer, and surgery	Culture	ND	33.0	[60]
12/9 male and 3 female/average 62/USA	*C. albicans, C. parapsilosis, C. glabrata, C. tropicalis*, and *C. dublinensis*	Mechanical ventilation, central venous catheter, ICU stay, and broad-spectrum antibiotics treatment	Culture and MALDI-TOF	ND	75.0	[61]

ICU: Intensive Care Unit; ND: no data; DM: diabetes mellitus.

**Table 3 jof-07-00556-t003:** Cases of COVID-19 and aspergillosis co-infection.

Number of Cases/Sex/Age (Years)/Country	Etiological Agent	Risk Factors	Diagnostic Method	Clinical Management	Fatality Rate (%)	Reference
1/female/72/China	*A. fumigatus*	Leflunomide for rheumatoid arthritis, Methylprednisolone, Tocilizumab, and glucocorticoid treatment	High-performance sequencing analysis	Caspofungin acetate	0.0	[23]
5/male:female ratio 2.2:1/Average 57/United Kingdom	*A. fumigatus*	Solid neoplasm	AspICU algorithm, BAL, culture, PCR, BDG, GM	Voriconazole caspofungin liposomal amphotericin B	53.0	[56]
14/ND/average 50.35/Mexico	*Aspergillus* spp., *A. fumigatus*, *A. flavus*, *A. niger*	Obesity, DM, hypertension, active smoker and HIV	Culture, MALDI-TOF, sGM	Voriconazole, anidulafungin	57.0	[62]
20/ND/elderly/USA and Spain	*A. fumigatus*	Severe immunosuppression due to hematological neoplasm or transplants, hypertensionlung disease, steroid therapy	BAL, culture, BDG	Voriconazole, posaconazole, liposomal mphotericin B	100.0	[63]
1/female/74/Netherlands	*A. fumigatus*	Hospitalization in the ICU	Culture, GM, BDG	Voriconazole, liposomal amphotericin B, caspofungin	100.0	[64]
1/male/71/Brazil	*A. penicillioides*	Hypertension DM, chronic kidney disease	Histopathology, GM, Confirmation by nucleotide sequencing	Post-mortem diagnosis	100.0	[65]
13/11 male and 2 female/average 54 to 78/Netherlands	*A. fumigatus*	Immunosuppression, ICU, VMI, prolonged use of corticosteroids treatment	BDG, GM, Fungal PCR targeting the *Cyp51A* gene	Voriconazole, caspofungin, liposomal amphotericin B	40.0 to 50.0	[66]
1/female/66/Australia	*A. section Fumigati*	Hypertension, smoking history, osteopenia, *Facklamia* *hominis* blood culture and *Escherichia coli* urine culture on admission	Non-bronchoscopic endotracheal aspirate with Gram staining	Voriconazole	0.0	[67]
1/male/ND/Austria	*Aspergillus* spp.	Chronic degenerative disease, neoplasia, immunosuppression	Autopsy	Post-mortem diagnosis	100.0	[68]
5/3 male and 2 female/average 69/Pakistan	*A. fumigatus*, *A. flavus*, *A. niger*	DM, high blood pressure	Culture, IgM, BDG	Voriconazole, liposomal amphotericin B	40.0	[69]
1/male/46/China	*A. fumigatus*	DM, stage 2 hypertension	Culture, MALDI-TOF	Voriconazole	0.0	[70]
1/female/87France	*Aspergillus* spp.	ND	GM, ELISA, Western blot, PCR	Voriconazole	100.0	[71]
1/female/58/Qatar	*A. niger*, *A. terreus*	Diabetic nephropathy, hypertension, hyperlipidemia, chronic hepatitis B infection, elderly patient	Culture	Anidulafungin, liposomal amphotericin B, voriconazole	100.0	[72]
1/male/74/France	*A. fumigatus*	Asymptomatic myelodysplastic syndrome (hypereosinophilia, with CD8+ T-cell lymphocytosis), Hashimoto’s thyroiditis, and hypertension	Culture, PCR, BDG, GM	No antifungal treatment was initiated due to the rapid and fatal course in the patient	100.0	[73]
1/female/42/Iran	*Aspergillus* spp.	Acute myeloid leukemia, DM	GM, ELISA for *Aspergillus*	Liposomal amphotericin B	100.0	[74]
2/1 male and 1 female/66 and 38 respectively/France	*Aspergillus* spp. *A. niger*	DM, obesity, hypertension, rheumatoid arthritis in methotrexate treatment	GM, PCR, BDG, Culture	ND	0.0	[75]
1/male/52/Denmark	*A. fumigatus sensu stricto*	DM, obesity, percutaneous coronary intervention	GM, MALDI-TOF	Voriconazole	0.0	[76]
10/8 male and 2 female/average between 51 and 76/Spain	*A. fumigatus*, *A. nidulans*	Hematological neoplasms, immunosuppression, DM, obesity, ICU COPD, Age > 65	Culture, MALDI-TOF, GM ELISA, AspICU algorithm	Corticosteroids, voriconazole, caspofungin, amphotericin B	70.0	[77]
1/male/66/Ireland	*A. fumigatus* with *TR34/L98H* mutation in *Cyp51A* gene	DM, hypertension, hyperlipidemia, obesity, grounds maintenance worker	BDG, GM	Liposomal amphotericin B	100.0	[78]
6/ND/average 55/France	*A. fumigatus*	Overweight, hypertension, DM, active smokers, COPD, Immunodepression	Culture, GMN, BDG, PCR	Voriconazole, caspofungin	57.2	[79]
7/5 male and 2 female/average 59.6 ± 15.21/Spain	*A. fumigatus*, *A. flavus, A. niger*	DM, obesity, sleep apnea, hypertension	PCR ITS1-5.8S-ITS2, GM	Itraconazole, liposomal amphotericin B	86.0	[80]
4/male/average 79/USA	*A. fumigatus*	COPD	AspICU algorithm, EORTC/MSG, culture, sGM	Voriconazole	100.0	[81]
9/ND/average 63.5/Germany	*A. fumigatus*	Hypertension	Microbiological follow-up tests	Echinocandins, voriconazole, fluconazole, addition of liposomal amphotericin B	13.0	[82]
1/male/56/France	*A. fumigatus* with TR34/L98H mutation in Cyp51A gene	DM, hypertension, hyperlipidemia, obesity	Tracheal aspirate, Cx + PCREUCAST, Pan-azole-resistance, autopsy was not performed	No antifungal treatment was given because patient died before results were obtained	100.0	[83]
1/male/73/Argentina	*A. s*ection *Fumigati*	Pulmonary embolism, and thrombophlebitis	GM, Pan-fungal nested PCR of 18S-rDNA	Voriconazole, liposomal amphotericin B, fluconazole	0.0	[84]
1/male/73/USA	*Aspergillus* spp., *A. flavus*	Hypertension	TAC, GM, culture	Voriconazole	0.0	[85]
1/male/85/Argentina	*A. flavus*	Hypertension	MALDI-TOF	Anidulafungin, voriconazole	100.0	[86]
1/female/55/Spain	*A. fumigatus*	Hypertension active smoker, liver hemangiomas, kidney transplant recipient	Sputum, culture, GM, BDG	Isavuconazole	0.0	[87]
8/6 male and 2 female/52 and 74 respectively/Spain	*A. fumigatus*, *A. terreus*, *A. awamori*, *A. citrinoterreus*, *A. lentulus*	Hypertension, obesity, asthma, kidney transplant recipient	Culture, sGM, PCR	Voriconazole, isavuconazole, liposomal amphotericin B	100.0	[88]
1/male/80/France	*A. flavus*	Removed thyroid cancer	Tracheal aspirate culture	Voriconazole, isavuconazole	100.0	[89]
30/24 male and 6 female/between 38 and 86 respectively/Germany, France, Netherlands, Belgium, Italy, Austria	*A. fumigatus*, *Aspergillus* spp. *A. flavus*	Obesity, DM, hypertension, chronic kidney disease, hyperlipidemia	Culture + GM, sGM, PCR	Voriconazole, isavuconazole, caspofungin, liposomal amphotericin B	50.0	[90]
6/ND/ND/United Kingdom	*A. fumigatus*	COVID-19 requiring hospitalization in the ICU	Culture, Microscopy, GM, BDG, PCR	ND	ND	[91]
10/ND/average 62/Netherlands	*Aspergillus* spp.		GM, PCR, Culture	ND	ND	[92]

BAL: Bronchoalveolar lavage; BDG: (1–3)-β-D-glucan; GM: Galactomannan; sGM: Serum galactomannan; ICU: Intensive Care Unit; COPD: Chronic obstructive pulmonary disease; DM: Diabetes Mellitus; ND: no Data.

**Table 4 jof-07-00556-t004:** Cases of COVID-19 and mucormycosis co-infection.

Number of Cases/Sex/Age (Years)/Country	Etiological Agent	Risk Factors	Diagnostic Method	Clinical Management	Fatality Rate (%)	Reference
1/male/60/India	Unidentified	DM, glucocorticoid treatment, and broad-spectrum antibiotics treatment	Clinical and suggestive MRI	Amphotericin B	0.0	[94]
1/male/86/Brazil	Unidentified	Glucocorticoid treatment and broad-spectrum antibiotics treatment	Histopathology, Gastric ulcer biopsy	ND	0.0	[95]
1/female/33/USA	Unidentified	Uncontrolled DM	Clinical and suggestive MRI	Amphotericin B and sinonasal debridement	0.0	[93]
1/male/55/India	*R. microsporus*	DM, glucocorticoid treatment, broad-spectrum antibiotics treatment, systemic high blood pressure, end-stage kidney disease, ischemic cardiomyopathy	Sputum sample culture	Liposomal amphotericin B, upper right lobectomy	100.0	[96]
1/male/49/USA	*Rhizopus* spp.	Glucocorticoid treatment, broad-spectrum antibiotics treatment	Histopathology, Right upper lobe biopsy	Amphotericin B	0.0	[97]
1/male/24/Mexico	*Lichteimia (Absidia)* spp.	Uncontrolled DM, diabetic ketoacidosis	Culture	Amphotericin B	0.0	[98]
1/male/66/Italy	*Rhizopus* spp.	Broad-spectrum antibiotics treatment	Bronchial aspirate culture	Liposomal amphotericin B, isavuconazole, thoracocentesis	0.0	[99]
1/male/53/Austria	*R. microsporus*	Neoplasia, glucocorticoids treatment	dPCR and sequencing, complete microscopic autopsy of lung tissue	ND	0.0	[100]
6/male/average 60.5/India	2-Unidentified and 4-*Mucor* spp.	DM, diabetic ketoacidosis, glucocorticoid treatment, and uncontrolled DM	Culture and histopathology	FESS and amphotericin B	100.0	[101]

DM: Diabetes mellitus, ND: No data; FESS: Functional endoscopic sinus surgery.

**Table 5 jof-07-00556-t005:** Cases of COVID-19 and endemic mycosis co-infection.

Number of Cases/Sex/Age (Years)/Country	Etiological Agent	Risk Factors	Diagnostic Method	Clinical Management	Fatality Rate (%)	Reference
1/female/48/USA	*C. immitis*	Heart failure, lived in an endemic area: Bakersfield, California	IgM and IgG by immunodiffusion assay with complement-fixation titers of 1:2	Fluconazole	0.0	[102]
1/male/48/USA	*C. immitis*	Uncontrolled DM, lives in endemic area: California	Positive serology for *Coccidioides* spp. with complement-fixation titers of 1:32	ND	0.0	[103]
1/female/43/Brazil	*H. capsulatum*	HIV infection with TCD4+ lymphocyte count of 113 cells/mm^3^, cocaine use, lived in an endemic area: Rio Grande, Brazil	Gomori-Grocott staining of expectoration, *H. capsulatum* urinary antigen	Itraconazole	0.0	[104]
1/male/43/Argentina	*H. capsulatum*	HIV infection with TCD4+ lymphocyte count of 16.3 cells/mm^3^,lived in an endemic area: Buenos Aires, Argentina	Giemsa staining, Blood culture and skin biopsy culture	Amphotericin B deoxycholate, itraconazole	0.0	[105]
1/female/36/Argentina	*H. capsulatum*	HIV infection with TCD4+ lymphocyte count of 3 cells/mm^3^, drug use: marijuana and cocaine, lived in endemic area: Buenos Aires, Argentina	Wright and Giemsa staining of expectoration, Histoplasma serum and urinary antigen	Amphotericin B deoxycholate, itraconazole	0.0	[106]

ND: no data; DM: diabetes mellitus; HIV: Human Immunodeficiency Virus.

## Data Availability

Not applicable.

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
