# Peer review of "Epidemiology of Systemic Mycoses in the COVID-19 Pandemic"

_jof, 2021, doi:10.3390/jof7070556_

Round 1

Reviewer 1 Report

The authors have made the necessary changes.

One last minor correction:

  1. Future challenges of systemic mycoses co-infections and COVID-19, rows 44-45:

Remove “Pfaller MA. Molecular approaches to diagnosing and managing infectious diseases: practicality and costs”.

After this small correction, there is no need to re-send the article for review.

Author Response

Open Review 1

English language and style

( ) Extensive editing of English language and style required
( ) Moderate English changes required
(x) English language and style are fine/minor spell check required
( ) I don't feel qualified to judge about the English language and style

Is the work a significant contribution to the field?

Is the work well organized and comprehensively described?

Is the work scientifically sound and not misleading?

Are there appropriate and adequate references to related and previous work?

Is the English used correct and readable?

Comments and Suggestions for Authors

The authors have made the necessary changes.

One last minor correction:

  1. Future challenges of systemic mycoses co-infections and COVID-19, rows 44-45:

Remove “Pfaller MA. Molecular approaches to diagnosing and managing infectious diseases: practicality and costs”.

The sentence “Pfaller MA. Molecular approaches to diagnosing and managing infectious diseases: practicality and costs”, was deleted in L44-L45, page 30

After this small correction, there is no need to re-send the article for review.

Reviewer 2 Report

Frías-De-León et al. conducted a literature search on the subject (January 2020 - February 2021) in PubMed, Embase, Cochrane Library, and LILACS without language restrictions. Eighty nine publications were found on co-infection by COVID-19 and pneumocystosis, candidiasis, aspergillosis, mucormycosis, coccidioidomycosis, or histoplasmosis. The study provides important information but also contains limitations that must be made explicit in the article (i.e. they are not official data and the number of reports depends on publications in the period). However, there are other points that need to be revised:

1- Tables - authors have to be sure about data formatting. The current format is not suitable.
2- It would be important to include the country of publication.
3- Page 29: "three of COVID-19 and histoplasmosis in Rio Grande, Brazil, and Buenos Aires, Argentina". In Brazil there are two states: Rio Grande do Norte and Rio Grande do Sul, which are geographically opposite. Please include the correct information.

Author Response

Open Review 2

English language and style

( ) Extensive editing of English language and style required
( ) Moderate English changes required
( ) English language and style are fine/minor spell check required
(x) I don't feel qualified to judge about the English language and style

Is the work a significant contribution to the field?

Is the work well organized and comprehensively described?

Is the work scientifically sound and not misleading?

Are there appropriate and adequate references to related and previous work?

Is the English used correct and readable?

Comments and Suggestions for Authors

Frías-De-León et al. conducted a literature search on the subject (January 2020 - February 2021) in PubMed, Embase, Cochrane Library, and LILACS without language restrictions. Eighty nine publications were found on co-infection by COVID-19 and pneumocystosis, candidiasis, aspergillosis, mucormycosis, coccidioidomycosis, or histoplasmosis. The study provides important information but also contains limitations that must be made explicit in the article (i.e. they are not official data and the number of reports depends on publications in the period). However, there are other points that need to be revised:

  • Tables - authors have to be sure about data formatting. The current format is not suitable.

The format is the one established in the norms for authors.

2- It would be important to include the country of publication.

      The country was included in the first column of the tables3- Page 29: "three of COVID-19 and histoplasmosis in Rio Grande, Brazil, and Buenos Aires, Argentina". In Brazil there are two states: Rio Grande do Norte and Rio Grande do Sul, which are geographically opposite. Please include the correct information.

      It was clarified that it was Rio Grande do Sul

Submission Date

04 June 2021

Date of this review

19 Jun 2021 01:09:02

Open Review 2

English language and style

( ) Extensive editing of English language and style required
( ) Moderate English changes required
( ) English language and style are fine/minor spell check required
(x) I don't feel qualified to judge about the English language and style

Is the work a significant contribution to the field?

Is the work well organized and comprehensively described?

Is the work scientifically sound and not misleading?

Are there appropriate and adequate references to related and previous work?

Is the English used correct and readable?

Comments and Suggestions for Authors

Frías-De-León et al. conducted a literature search on the subject (January 2020 - February 2021) in PubMed, Embase, Cochrane Library, and LILACS without language restrictions. Eighty nine publications were found on co-infection by COVID-19 and pneumocystosis, candidiasis, aspergillosis, mucormycosis, coccidioidomycosis, or histoplasmosis. The study provides important information but also contains limitations that must be made explicit in the article (i.e. they are not official data and the number of reports depends on publications in the period). However, there are other points that need to be revised:

  • Tables - authors have to be sure about data formatting. The current format is not suitable.

The format is the one established in the norms for authors.

2- It would be important to include the country of publication.

      The country was included in the first column of the tables

3- Page 29: "three of COVID-19 and histoplasmosis in Rio Grande, Brazil, and Buenos Aires, Argentina". In Brazil there are two states: Rio Grande do Norte and Rio Grande do Sul, which are geographically opposite. Please include the correct information.

      It was clarified that it was Rio Grande do Sul

Submission Date

Open Review 2

English language and style

( ) Extensive editing of English language and style required
( ) Moderate English changes required
( ) English language and style are fine/minor spell check required
(x) I don't feel qualified to judge about the English language and style

Is the work a significant contribution to the field?

Is the work well organized and comprehensively described?

Is the work scientifically sound and not misleading?

Are there appropriate and adequate references to related and previous work?

Is the English used correct and readable?

Comments and Suggestions for Authors

Frías-De-León et al. conducted a literature search on the subject (January 2020 - February 2021) in PubMed, Embase, Cochrane Library, and LILACS without language restrictions. Eighty nine publications were found on co-infection by COVID-19 and pneumocystosis, candidiasis, aspergillosis, mucormycosis, coccidioidomycosis, or histoplasmosis. The study provides important information but also contains limitations that must be made explicit in the article (i.e. they are not official data and the number of reports depends on publications in the period). However, there are other points that need to be revised:

  • Tables - authors have to be sure about data formatting. The current format is not suitable.

The format is the one established in the norms for authors.

2- It would be important to include the country of publication.

      The country was included in the first column of the tables3- Page 29: "three of COVID-19 and histoplasmosis in Rio Grande, Brazil, and Buenos Aires, Argentina". In Brazil there are two states: Rio Grande do Norte and Rio Grande do Sul, which are geographically opposite. Please include the correct information.

      It was clarified that it was Rio Grande do Sul

04 June 2021

Date of this review

19 Jun 2021 01:09:02

Round 2

Reviewer 2 Report

There are some typos in the text. for example in Table 1 - Correct: France and not Fracia or Francia. The article needs proofreading to eliminate some errors in the text.

This manuscript is a resubmission of an earlier submission. The following is a list of the peer review reports and author responses from that submission.

Round 1

Reviewer 1 Report

Paper still requires modifications:

1- The authors should incorporate references to ALL PARAGRAPHS discussing the association between COVID and specific fungal infections

2- The authors stated that lymphopenia increases colonization and infection by fungi. I don't think we have data to support their hypothesis that lymphopenia per se increases the rate of colonization... 

3- The authors included nystatin as an alternatives used to treat Candida invasive infections. This is an antifungal formulation with no impact in invasive candidiasis !! This is a formulation that may be used for treating oral candidiasis but not systemic candidiasis.

4- The authors presented lots of data on diagnosis and treatment of fungal infections without substantial reflections and criticism about their accuracy and effectiveness. 

Author Response

  • The authors should incorporate references to ALL PARAGRAPHS discussing the association between COVID and specific fungal infections.

In each of the paragraphs that discuss the association between COVID and specific fungal infections, the references were incorporated.

Pages 5, 8, 18, 26, 29

  • The authors stated that lymphopenia increases colonization and infection by fungi. I don't think we have data to support their hypothesis that lymphopenia per se increases the rate of colonization... 

This paragraph has been corrected, as there is indeed no data to support the hypothesis that lymphopenia per se increases the colonization rate

L93-L96, page 3.

  • The authors included nystatin as an alternatives used to treat Candida invasive infections. This is an antifungal formulation with no impact in invasive candidiasis !! This is a formulation that may be used for treating oral candidiasis but not systemic candidiasis.

Indeed, according to the literature reviewed, nystatin was used for oropharyngeal candidiasis, therefore, this paragraph was corrected.

L15-L16, page 8

  • The authors presented lots of data on diagnosis and treatment of fungal infections without substantial reflections and criticism about their accuracy and effectiveness. 

Substantial and critical reflections on the precision and efficacy of the large amount of data on the diagnosis and treatment of fungal infections were added in two paragraphs within numeral 8.

L38-L45, page 32,

L58-L62, page 33

Reviewer 2 Report

The authors addressed the provided comments in an appropriate way.

Author Response

The authors addressed the provided comments in an appropriate way.

. No change to the text of the article was suggested

Reviewer 3 Report

Materials and methods

Figure 1:

- in the abstract 94 articles are mentioned (in part 6. Mucormycosis: 14 publications but 9 in table), but in the PRISMA diagram n=89; please correct in abstract (89 instead of 94) and in part 6. Mucormycosis and COVID-19 (9 instead of 14).

Author Response

Figure 1:

In the abstract 94 articles are mentioned (in part 6Mucormycosis14 publications but 9 in table), but in the PRISMA diagram n=89; please correct in abstract (89 instead of 94) and in part 6. Mucormycosis and COVID-19 (9 instead of 14).

89 was placed instead of 94 in the abstract, L31, page 1

9 was placed instead of 14 in mucormycosis, L2, page 26